# Alternating Minimization for Regression Problems with Vector-valued Outputs

**Prateek Jain**
Microsoft Research, INDIA
prajain@microsoft.com

**Ambuj Tewari**
University of Michigan, Ann Arbor, USA
tewaria@umich.edu

## Abstract

In regression problems involving vector-valued outputs (or equivalently, multiple responses), it is well known that the maximum likelihood estimator (MLE), which takes noise covariance structure into account, can be significantly more accurate than the ordinary least squares (OLS) estimator. However, existing literature compares OLS and MLE in terms of their asymptotic, not finite sample, guarantees. More crucially, computing the MLE in general requires solving a non-convex optimization problem and is not known to be efficiently solvable. We provide finite sample upper and lower bounds on the estimation error of OLS and MLE, in two popular models: a) Pooled model, b) Seemingly Unrelated Regression (SUR) model. We provide precise instances where the MLE is significantly more accurate than OLS. Furthermore, for both models, we show that the output of a computationally efficient alternating minimization procedure enjoys the same performance guarantee as MLE, up to universal constants. Finally, we show that for high-dimensional settings as well, the alternating minimization procedure leads to significantly more accurate solutions than the corresponding OLS solutions but with error bound that depends only logarithmically on the data dimensionality.

## 1 Introduction

Regression problems with vector-valued (or, equivalently, multiple) response variables – where we want to predict multiple responses based on a set of predictor variables– is a classical problem that arises in a wide variety of fields such as economics [1, 2, 3], and genomics [4]. In such problems, it is natural to assume that the noise, or error, terms in the underlying linear regression model are correlated across the response variables. For example, in multi-task learning, the errors in different task outputs can be heavily correlated due to similarity of the tasks.

Regression with multiple responses is a classical topic. Textbooks in statistics [5, 6] and econometrics [7] cover it in detail and illustrate practical applications. [2] and [3] provide recent overviews of the Seemingly Unrelated Regressions (SUR) model and the associated estimation procedures. It is well known that for SUR models, the standard Ordinary Least Squares (OLS) estimator may not be (asymptotically) efficient (i.e., may not achieve the Cramer-Rao lower bound on the asymptotic variance) and that efficiency can be gained by using an estimator that exploits noise correlations [3] such as the Maximum Likelihood Estimator (MLE). The two well-known exceptions to this underperformance of OLS are: when the noise across tasks is uncorrelated and when the regressors are shared across tasks. The later is the well-known multivariate regression (MR) setting (see [5, Chapter 6]). However, there are at least two limitations of the existing MLE literature in this context.

First, despite being a classical and widely studied problem, little attention has been paid to the fact that MLE involves solving a *non-convex* optimization problem in general and is not known to be efficiently solvable. For example, a standard text in econometrics [7, p. 298, footnote 15], when discussing the SUR model, says, "*We note, this procedure [i.e., AltMin] produces the MLE when it converges, but it is not guaranteed to converge, nor is it assured that there is a unique MLE.*" The

text also cites [8] to claim that "*if the [AltMin] iteration converges, it reaches the MLE*" but the result [8, Theorem 1] itself only claims that "*the iterative procedure always converges to a solution of the first-order maximizing conditions*" and not necessarily to "*the **absolute** maximum of the likelihood function*" (emphasis on the word "absolute" is in the original text).

Second, improvement claims for MLE over OLS are based on *asymptotic* efficiency comparisons [7, Chapter 10] that are valid only in the limit as the sample size goes to infinity. Little is known about the estimation error with a finite number of samples. When discussing the failure of AltMin to converge even after 1,000 iterations, the text [7] says that the "problem with this application may be the very small sample size, 17 observations". This is consistent with our theoretical results that guarantee error bounds for AltMin once the sample size is large enough (in a quantifiable way).

The main contribution of this paper is quantifying, via *finite sample bounds*, the improvement in estimation error resulting from joint estimation of the regression coefficients and the noise covariance. Our approach is firmly rooted in the statistical learning theory tradition: we pay attention to *efficient computation* and use *concentration inequalities*, rather than limit theorems, to derive finite sample guarantees. In order to have a computationally efficient approach, we adopt an alternating minimization (AltMin) procedure that alternatingly estimates the regression coefficients and the noise covariance matrix, while keeping the other unknown fixed. While both of the individual problems are "easy" to solve and can be implemented efficiently, the general problem is still non-convex and such a procedure might lead to local optima. Whereas practitioners have long recognized that AltMin works well for such problems [1, Chapter 5], we are not aware of any provable guarantees for it in the setting of multiple response regression.

We consider two widely-used vector-output models, namely the Pooled model (Section 2) and the Seemingly Unrelated Regression (SUR) model (Section 3). For both models, we show that the estimation error of AltMin matches the MLE solution's error up to log factors. Moreover, we show that in general, the error bounds of MLE (and AltMin) are significantly better than that of OLS. To derive our finite sample guarantees, we rely on concentration inequalities from random matrix theory. For AltMin, our proof exploits a *virtuous circle*: better estimation of the regression coefficients helps covariance estimation and vice-versa. As a result, we are able to show that the both parameter estimation errors reduce by at least a constant factor in each iteration of AltMin.

**Illustrative Example.** To whet the reader's appetite for what follows, we consider here a simple regression problem with two responses: $y_{i,1} = X_{i,1}^\top \mathbf{w}_* + \eta_i, \quad y_{i,2} = X_{i,2}^\top \mathbf{w}_* + \eta_i, \ 1 \leq i \leq n$, where $X_{i,1}, X_{i,2} \in \mathbb{R}^d$ are drawn i.i.d. from the spherical normal distribution. The coefficient vector $\mathbf{w}_*$ is shared across the two problems, which holds true in the pooled model studied in Section 2. Later in Section 3, we also consider the SUR model that allows for different coefficient vectors across problems. More importantly, notice that the i.i.d. noise, $\eta_i$ (say, it is standard Gaussian) is *shared* across the two problems. If we estimate $\mathbf{w}_*$ using OLS:

$$\mathbf{w}_{OLS} = \arg\min_{\mathbf{w}} \frac{1}{n} \sum_{i=1}^n (y_{i,1} - X_{i,1}^\top \mathbf{w})^2 + \frac{1}{n} \sum_{i=1}^n (y_{i,1} - X_{i,2}^\top \mathbf{w})^2$$

then we will have $\|\mathbf{w}_{OLS} - \mathbf{w}_*\|_2 = \Omega(1/\sqrt{n})$. However, subtracting the two equations gives: $y_{i,1} - y_{i,2} = (X_{i,1} - X_{i,2})^\top \mathbf{w}_*, \ 1 \leq i \leq n$. That is, as soon as we have $n \geq d$ samples, we will recover $\mathbf{w}_*$ *exactly* by solving the above system of linear equations!

Our toy example motivates the fundamental question that this paper answers: how much can we improve OLS by exploiting noise correlations? Let us make the example more realistic by assuming the model: $\mathbf{y}_i = X_i \mathbf{w}_* + \boldsymbol{\eta}_i, \ 1 \leq i \leq n$, where $\mathbf{y}_i \in \mathbb{R}^m$ is a vector of $m$ responses, each element of $X_i \in \mathbb{R}^{m \times d}$ is sampled i.i.d. from the standard Gaussian and noise vector $\boldsymbol{\eta}_i$ is drawn from $\mathcal{N}(0, \Sigma_*)$. A corollary of the main result in Section 2 shows that MLE (see (3)) improves upon the OLS parameter error bound by a factor of $\text{Error}_{OLS}/\text{Error}_{MLE} = \text{tr}(\Sigma_*) \text{tr}(\Sigma_*^{-1})/m^2$. This factor can easily be seen to be larger than 1 by using Cauchy-Schwarz inequality: $\text{tr}(\Sigma_*) \text{tr}(\Sigma_*^{-1})/m^2 = (\sum_j \lambda_j)(\sum_j 1/\lambda_j)/m^2 \geq (\sum_j \sqrt{\lambda_j} \cdot 1/\sqrt{\lambda_j})^2/m^2 = 1$, where $\lambda_j$ be the $j$-th largest eigenvalue of $\Sigma_*$. The inequality is tight when $\sqrt{\lambda_j} = c/\sqrt{\lambda_j}$ for some constant $c$. That is, when $\Sigma_* = cI$ which holds true iff the noise in each response is mutually independent and has same variance. The more $\Sigma_*$ departs from being $c \cdot I$, the larger the improvement factor. For example, consider $m = 2$ case again, but rather than $\eta_{i,1} = \eta_{i,2}$, we have highly correlated $[\eta_{i,1}, \eta_{i,2}]$ with covariance matrix $\Sigma_* = \begin{bmatrix} 1 & 1-\epsilon \\ 1-\epsilon & 1 \end{bmatrix}$. So, $\Sigma_*^{-1} = \frac{1}{2\epsilon - \epsilon^2} \begin{bmatrix} 1 & -1+\epsilon \\ -1+\epsilon & 1 \end{bmatrix}$. The improvement factor becomes

$\text{tr}(\Sigma_*)\text{tr}(\Sigma_*^{-1})/m^2 = 1/(2\epsilon - \epsilon^2)$ which blows up to $\infty$ as $\epsilon \to 0$. As mentioned earlier, we show a similar improvement for the output of a computationally efficient AltMin procedure.

**Related Works.** Vector-output regression problems are also studied in the context of multi-task learning. Following the terminology introduced by [9], we can classify this literature as exploiting *task structure* (shared structure in the regression coefficients) or *output structure* (correlation in noise across tasks) or both. The large body of work [10, 11, 12] on structured sparsity regularization and on using (reproducing) kernels for multi-task learning [13, 14], falls mostly into the former category. In this body of work, problem formulations are often convex and efficient learning algorithms with finite sample guarantees can be derived. Our focus in this paper, however, is on methods that exploit *noise correlation*. Rai et al. [9] summarize the relevant multitask literature on exploiting output structure and provide novel results by exploiting both task and output structure simultaneously. Neither they, nor the work they cite, provide any finite sample guarantees for the iterative procedures employed. The same comment applies to work in high-dimensional settings on learning structured sparsity as well as output structure via joint regularization of regression coefficients and noise covariance matrix [15, 16, 17]. We hope that techniques developed in this paper pave the way for studying such joint regularization problems involving non-convex objectives.

Recent results have shown that alternating minimization leads to exact parameter recovery in certain observation models such as matrix completion [18], and dictionary learning [19]. However, most of the existing results are concerned with exact parameter estimation and their techniques do not apply to our problems. In contrast, we provide better statistical rates by exploiting the hidden noise covariance matrix. To the best of our knowledge, ours is first such result for AltMin in the statistical setting where AltMin leads to dramatic improvement in the error rates.

**Notations.** Vectors are in general represented using bold-face letters, e.g. $\mathbf{w}$. Matrices are represented by capital letters, e.g. $W$. For data matrix $X \in \mathbb{R}^{m \times d}$, $X^j \in \mathbb{R}^{1 \times d}$ represents the $j$-th row of $X$. Throughout the paper, $\Sigma_X = \mathbb{E}_X[X^T X]$ is the covariance of the data matrix and $\Sigma_*$ denotes the covariance of the noise matrix. $\lambda_j(\Sigma)$ denotes the $j$-largest eigenvalue of $\Sigma \in \mathbb{R}^{m \times m}$. That is, $\lambda_{\max}(\Sigma) = \lambda_1(\Sigma) \geq \lambda_2(\Sigma) \cdots \geq \lambda_m(\Sigma) = \lambda_{\min}(\Sigma)$ are the eigenvalues of $\Sigma$. Universal constants denoted by "$C$" can take different values in different steps. $\|A\|_2 = \max_{\mathbf{u}, \|\mathbf{u}\|_2 = 1} \|A\mathbf{u}\|_2$ denotes the spectral norm of $A$, while $\|A\|_F$ denotes the Frobenius norm of $A$. Following Matlab notation, $\text{diag}(A)$ represents the vector of diagonal entries of $A$.

## 2 The Pooled Model

We first consider a pooled model where a single coefficient vector is used across all data points and tasks (hence the name "pooled" [7]). It may seem that the model is very restrictive compared to the MR and SUR models. However, as we show later, by vectorizing the coefficient matrices, both MR and SUR models can be thought of as special cases of the pooled model. Moreover, the pooled model is in itself interesting for several applications, such as query-document rankings. For example, the ranking method of [20] is equivalent to OLS estimation under the pooled model.

Let $\mathcal{D} = \{(X_1, \mathbf{y}_1), \ldots, (X_n, \mathbf{y}_n)\}$ where the $i$-th data point $X_i \in \mathbb{R}^{m \times d}$, and its output $\mathbf{y}_i \in \mathbb{R}^m$. $m$ denotes the number of "tasks" and $d$ is the "data" dimensionality. Given $\mathcal{D}$, the goal is to learn weights $\mathbf{w} \in \mathbb{R}^d$ s.t. $X\mathbf{w} \approx \mathbf{y}$ for a novel data point $X$ and the target output $\mathbf{y}$. We assume that the data is generated according to the following model:

$$\mathbf{y}_i = X_i \mathbf{w}_* + \boldsymbol{\eta}_i,\ 1 \leq i \leq n, \tag{1}$$

where $\mathbf{w}_* \in \mathbb{R}^d$ is the optimal parameter vector we wish to learn, data points $X_i \overset{i.i.d.}{\sim} \mathcal{P}_X, 1 \leq i \leq n$ and the noise vectors $\boldsymbol{\eta}_i \overset{i.i.d.}{\sim} \mathcal{N}(0, \Sigma_*)$ are sampled independent of $X_i$'s.

A straightforward approach to estimating $\mathbf{w}_*$ is to ignore correlation in the noise vector $\boldsymbol{\eta}_i$ and treat the problem as a large regression problem with $m \cdot n$ examples. That is, perform the *Ordinary Least Squares* (OLS) procedure:

$$\mathbf{w}_{OLS} = \arg\min_{\mathbf{w}} \frac{1}{n} \sum_{i=1}^{n} \|\mathbf{y}_i - X_i \mathbf{w}\|_2^2. \tag{2}$$

It is easy to see that the above solution is "consistent". That is, for $n \to \infty$, we have $E_{X \sim \mathcal{P}_X}[\|X\mathbf{w} - X\mathbf{w}_*\|_2^2] \to 0$. However, intuitively, by using the noise correlations, one should be able to obtain significantly more accurate solution for finite $n$.

**Algorithm 1** AltMin-Pooled: Alternating Minimization for the Pooled Model

---
**Require:** $\mathcal{D} = \{(X_1, \mathbf{y}_1) \ldots (X_{2nT}, \mathbf{y}_{2nT})\}$, Number of iterations: $T$
 1: Randomly partition $\mathcal{D} = \{\mathcal{D}_0^\Sigma, \mathcal{D}_0^\mathbf{w}, \mathcal{D}_1^\Sigma, \mathcal{D}_1^\mathbf{w}, \ldots, \mathcal{D}_T^\Sigma, \mathcal{D}_T^\mathbf{w}\}$, where $|\mathcal{D}_t^\mathbf{w}| = |\mathcal{D}_t^\Sigma| = n, \forall t$
 2: Initialize $\mathbf{w}_0 = 0$
 3: **for** $t = 0, \ldots, T-1$ **do**
 4:    Covariance Estimation: $\widehat{\Sigma}_t = \frac{1}{n} \sum_{i \in \mathcal{D}_t^\Sigma} (\mathbf{y}_i - X_i \mathbf{w}_t)(\mathbf{y}_i - X_i \mathbf{w}_t)^T$
 5:    Least-squares Solution: $\mathbf{w}_{t+1} = \arg\min_\mathbf{w} \frac{1}{n} \sum_{i \in \mathcal{D}_t^\mathbf{w}} \|\widehat{\Sigma}_t^{-\frac{1}{2}}(\mathbf{y}_i - X_i \mathbf{w})\|_2^2$
 6: **end for**
 7: **Output**: $\mathbf{w}_T$

---

Ideally, if $\Sigma_*$ was known, we would like to estimate $\mathbf{w}_*$ by decorrelating the noise[1]. That is,

$$\mathbf{w}_{MLE} = \arg\min_\mathbf{w} \frac{1}{n} \sum_{i=1}^n \|\Sigma_*^{-\frac{1}{2}}(\mathbf{y}_i - X_i \mathbf{w})\|_2^2. \tag{3}$$

However, $\Sigma_*$ is not known apriori and in general can only be estimated if $\mathbf{w}_*$ is known. To avoid this circular requirement, we can jointly estimate $(\mathbf{w}_*, \Sigma_*)$ by maximizing the joint likelihood. The joint maximum likelihood estimation (MLE) problem for $(\mathbf{w}, \Sigma)$ is given by:

$$(\widehat{\mathbf{w}}, \widehat{\Sigma}) = \arg\max_{\mathbf{w}, \Sigma \succeq 0} -\log|\Sigma| - \frac{1}{n} \sum_{i=1}^n (\mathbf{y}_i - X_i \mathbf{w})^T \Sigma^{-1} (\mathbf{y}_i - X_i \mathbf{w}). \tag{4}$$

The problem above is *non-convex* in $(\Sigma, \mathbf{w})$ jointly, and hence standard convex optimization techniques do not apply to the problem. A straightforward heuristic approach is to use alternating minimization (AltMin) where we alternately solve for $\widehat{\mathbf{w}}$ (and $\widehat{\Sigma}$) while keeping $\widehat{\Sigma}$ (and $\widehat{\mathbf{w}}$) fixed. Note that, each of the above mentioned individual problems are fairly straightforward and can be solved efficiently (see Steps 4, 5 of Algorithm 1). Despite its simplicity and availability of optimal solutions at each iteration, AltMin need not converge to a global optima of the joint problem. Below, we show that despite non-convexity of (4), we can still show that the the AltMin procedure has a matching *error bound* when compared to the optimal MLE solution.

Specifically, we analyze Algorithm 1 which is just the standard AltMin method but uses fresh samples $(\mathbf{y}, X)$ for each of the covariance estimation and the least squares step. Practical systems do not perform such re-sampling, but fresh samples at every iteration ensure that errors do not get correlated in adversarial fashion and allows us to use standard concentration bounds. Moreover, since we show convergence at a geometric rate, the number of iterations is not large and hence the sample complexity does not increase by a significant factor.

To prove our convergence results, we require the probability distribution $\mathcal{P}_X$ to be a sub-Gaussian distribution with the sub-Gaussian norm ($\|X\|_{\psi_2}$) defined as:

**Definition 1.** *Let $X \in \mathbb{R}^{m \times d}$ be a random variable (R.V.) with distribution $\mathcal{P}_X$. Then, the sub-Gaussian norm of $X$ is given by:*

$$\|X\|_{\psi_2} = \max_{\substack{\mathbf{u}, \|\mathbf{u}\|_2=1 \\ \mathbf{v}, \|\mathbf{v}\|_2=1}} \|\mathbf{v}^T \Sigma_{Xu}^{-\frac{1}{2}} X^T \mathbf{u}\|_{\psi_2}, \ \ where, \ \Sigma_{Xu} = \mathbb{E}_{X \sim \mathcal{P}_X}[X^T \mathbf{u}\mathbf{u}^T X].$$

*Sub-Gaussian norm of a univariate variable $Q$ is defined as: $\|Q\|_{\psi_2} = \max_{p \geq 1} \frac{1}{\sqrt{p}} \cdot \mathbb{E}[|Q|^p]^{\frac{1}{p}}$. If $\Sigma_{Xu}$ is not invertible for any fixed $u$ then, we define $\|X\|_{\psi_2} = \infty$*

We pre-multiply $X^T \mathbf{u}$ by $\Sigma_{Xu}^{-\frac{1}{2}}$ for normalization, so that for Gaussian $X$, $\|X\|_{\psi_2} = 1$. For bounded variables $X$, s.t., each entry $|X_{ij}| \leq M$, we have: $\|X\|_{\psi_2} \leq M\sqrt{md} \cdot \max_{\mathbf{u}, \|\mathbf{u}\|_2=1} \|\Sigma_{Xu}^{-1}\|_2$.

**Theorem 2** (Result for Pooled Model)**.** *Let $X_i \overset{i.i.d.}{\sim} \mathcal{P}_X, 1 \leq i \leq n$ with sub-Gaussian norm $\|X_i\|_{\psi_2} < \infty$ and $\boldsymbol{\eta}_i \overset{i.i.d.}{\sim} \mathcal{N}(0, \Sigma_*)$ are independent of $X_i$'s. Let $\mathbf{w}_* \in \mathbb{R}^d$ be a fixed vector and*

$n \geq C \cdot (m + d)\|X\|_{\psi_2}$. *Then, the output* $\mathbf{w}_T$ *of Algorithm 1 satisfies (w.p.* $\geq 1 - \frac{T}{n^{10}}$):

$$\mathbb{E}_{X \sim \mathcal{P}_X} \left[ \|X(\mathbf{w}_T - \mathbf{w}_*)\|_2^2 \right] \leq \frac{Cd \log n}{n} \cdot \frac{1}{\lambda_{\min}^*} + \frac{\lambda_{\max}^*}{\lambda_{\min}^*} 2^{-T},$$

*where* $\lambda_{\min}^* = \lambda_{\min}(\Sigma_{X*})$, $\lambda_{\max}^* = \lambda_{\max}(\Sigma_{X*})$, *and* $\Sigma_{X*} = \mathbb{E}_{X \sim \mathcal{P}_X}[\Sigma_X^{-\frac{1}{2}} X^T \Sigma_*^{-1} X \Sigma_X^{-\frac{1}{2}}]$. *Also,* $\Sigma_X = E_{X \sim \mathcal{P}_X}[X^T X]$ *is the covariance of the regressors.*

**Remarks**: Using Theorem 15, we also have the following bound for the OLS solution:

$$\mathbb{E}_{X \sim \mathcal{P}_X} \left[ \|X(\mathbf{w}_{OLS} - \mathbf{w}_*)\|_2^2 \right] \leq \frac{C \cdot d \log n}{n} \cdot \|\Sigma_*\|_2.$$

The above bound for OLS can be shown to be tight as well (up to $\log n$ factor) by selecting each $X_i = \mathbf{u}_{\max}$; $\mathbf{u}_{\max}$ is the eigenvector of $\Sigma_*$ corresponding to $\lambda_{\max}(\Sigma_*)$. Now, it is easy to see that: $\frac{1}{\lambda_{\min}^*} \leq \|\Sigma_*\|_2$ (see Claim 17). Hence, our bound for AltMin (as well as MLE) is tighter than that of OLS. Sub-sections 2.1 and 2.2 demonstrates gains over OLS in several standard settings.

Our proof of the above theorem critically uses the following lemma which shows that a particular potential function drops (up to MLE error) geometrically at each step of the AltMin procedure.

**Lemma 3.** *Assume the notation of Theorem 2. Let* $\mathbf{w}_{t+1}$ *be the* $(t + 1)$-*th iterate of Algorithm 1. Then, the following holds w.p.* $\geq 1 - 1/n^{10}$:

$$\mathbb{E}_{X \sim \mathcal{P}_X} \left[ \|\Sigma_*^{-\frac{1}{2}} X(\mathbf{w}_{t+1} - \mathbf{w}_*)\|_2^2 \right] \leq \frac{2C \cdot d \log n}{n} + \frac{1}{2} \cdot \mathbb{E}_{X \sim \mathcal{P}_X} \left[ \|\Sigma_*^{-\frac{1}{2}} X(\mathbf{w}_t - \mathbf{w}_*)\|_2^2 \right].$$

See Appendix C for detailed proofs of both of the results given above.

## 2.1 Gaussian $X$: Independent Rows

We first consider a special case where each row of $X$ is sampled i.i.d. from a Gaussian distribution. That is,
$$X_i^j \stackrel{i.i.d.}{\sim} \mathcal{N}(0, \Lambda), \ \forall 1 \leq i \leq n, \ 1 \leq j \leq m,$$
where $\Lambda \succ 0 \in \mathbb{R}^{d \times d}$ is a covariance matrix and $\Sigma_X = \mathbb{E}_{X \sim \mathcal{P}_X}[X^T X] = m \cdot \Lambda$. Let $\Sigma_* = \sum_{j=1}^m \lambda_i(\Sigma_*) \mathbf{u}_i \mathbf{u}_i^T$ be the eigenvalue decomposition of $\Sigma_*$. Then,

$$\Sigma_{X*} = \mathbb{E}_{X \sim \mathcal{P}_X}[\Sigma_X^{-\frac{1}{2}} X^T \Sigma_*^{-1} X \Sigma_X^{-\frac{1}{2}}] = \sum_j \frac{\mathbb{E}_{X \sim \mathcal{P}_X}[\Sigma_X^{-\frac{1}{2}} X^T \mathbf{u}_i \mathbf{u}_i^T X \Sigma_X^{-\frac{1}{2}}]}{\lambda_i(\Sigma_*)} = \frac{\text{tr}(\Sigma_*^{-1})}{m} I_{d \times d}.$$

We now combine the above given observation with Theorem 2 to obtain our error bound for AltMin procedure. Using a slightly stronger version of Theorem 15, we can also obtain the error bound for the OLS (Ordinary Least Squares) solution as well the MLE solution.

**Corollary 4** (Result for Pooled Model, Gaussian Data, Independent Rows). *Let* $X_i$ *be sampled s.t. each row* $X_i^j \sim \mathcal{N}(0, \Lambda)$ *and* $\Lambda \succ 0$. *Also, let* $\mathbf{y}_i = X_i \mathbf{w}_* + \boldsymbol{\eta}_i$, *where* $\boldsymbol{\eta}_i \sim \mathcal{N}(0, \Sigma_*)$, $\Sigma_* \succ 0$. *Let* $n \geq C(m + d) \log(m + d)$. *Then, the OLS solution* (2) *and the MLE solution* (3) *has the following error bounds (w.p.* $\geq 1 - 1/n^{10}$):

$$\mathbb{E}_X[\|X(\mathbf{w}_{OLS} - \mathbf{w}_*)\|_2^2] \leq \frac{Cd \log n}{n} \cdot \frac{\text{tr}(\Sigma_*)}{m}, \ \ \mathbb{E}_X[\|X(\mathbf{w}_{MLE} - \mathbf{w}_*)\|_2^2] \leq \frac{Cd \log n}{n} \cdot \frac{m}{\text{tr}(\Sigma_*^{-1})}.$$

*Moreover, the output* $\mathbf{w}_T$ ($T = \log \frac{1}{\epsilon}$) *of Algorithm 1 satisfies (w.p.* $\geq 1 - T/n^{10}$):

$$\text{Error}_T = \mathbb{E}_{X \sim \mathcal{P}_X}[\|X(\mathbf{w}_T - \mathbf{w}_*)\|_2^2] \leq \frac{8Cd \log n}{n} \cdot \frac{m}{\text{tr}(\Sigma_*^{-1})} + \epsilon.$$

**Lower Bound for OLS and MLE:** We now show that the error bounds for both the OLS as well as the MLE solution stated above are in fact tight up to log-factors.

**Lemma 5.** *Let the assumptions of Corollary 4 hold. Then, we have (w.p.* $\geq 1 - 1/n^{10} - \exp(-d)$):

$$\mathbb{E}_X[\|X(\mathbf{w}_{OLS} - \mathbf{w}_*)\|_2^2] \geq \frac{Cd}{n} \cdot \frac{\text{tr}(\Sigma_*)}{m}, \ \ E_X[\|X(\mathbf{w}_{MLE} - \mathbf{w}_*)\|_2^2] \geq \frac{Cd}{n} \cdot \frac{m}{\text{tr}(\Sigma_*^{-1})},$$

*where* $C > 0$ *is a universal constant.*

**Remarks**: As mentioned in the introduction, $\frac{m}{\operatorname{tr}(\Sigma_*^{-1})} \leq \frac{\operatorname{tr}(\Sigma_*)}{m}$ and the gap becomes larger as $\Sigma_*$ moves away from $c \cdot I$. Hence, in the light of the above two lower-bound and upper-bound results, it is clear that AltMin (and MLE) solutions are significantly more accurate than OLS, especially for highly correlated noise vectors. This claim is also bore out from our simulation results (Figure 4).

## 2.2 Gaussian $X$: Dependent Rows

We now generalize the above given special case by removing the row-wise independence assumption. That is, $X = \Sigma_R^{\frac{1}{2}} Z \Lambda^{\frac{1}{2}}$, where $Z_{ij} \overset{i.i.d.}{\sim} \mathcal{N}(0,1) \ \forall i, j$ and $\Sigma_R \in \mathbb{R}^{m \times m}$, $\Lambda \in \mathbb{R}^{d \times d}$ are the row and the column correlation matrices, respectively. It is easy to see that (see Claim 18),

$$\Sigma_{X*} = \mathbb{E}_{X \sim \mathcal{P}_X}[\Sigma_X^{-\frac{1}{2}} X^T \Sigma_*^{-1} X \Sigma_X^{-\frac{1}{2}}] = \frac{\operatorname{tr}(\Sigma_R \Sigma_*^{-1})}{\operatorname{tr}(\Sigma_R)} \cdot I_{d \times d}, \text{ where } \Sigma_X = \operatorname{tr}(\Sigma_R) \cdot \Lambda.$$

Using Theorem 2 with Theorem 15 and (32) (with certain $A$, $B$) we obtain the following corollary.

**Corollary 6** (Result for Pooled Model, Gaussian Data, Dependent Rows). *Let $X_i$ be as defined above. Let $n \geq C(m + d) \log(m + d)$. Then the followings holds (w.p. $\geq 1 - T/n^{10}$):*

$$\mathbb{E}_{X \sim \mathcal{P}_X}[\|X(\mathbf{w}_T - \mathbf{w}_*)\|_2^2] \leq \frac{8Cd \log n}{n} \cdot \frac{m}{\operatorname{tr}(\Sigma_R \Sigma_*^{-1})} + \epsilon,$$

*where $\mathbf{w}_T$ is the output of Algorithm 1 with $T = \log \frac{1}{\epsilon}$.*

Similarly, bound for OLS is given by: $\mathbb{E}_{X \sim \mathcal{P}_X}[\|X(\mathbf{w}_{OLS} - \mathbf{w}_*)\|_2^2] \leq \frac{Cd \log n}{n} \cdot \frac{m \cdot \operatorname{tr}(\Sigma_R \Sigma_*)}{\operatorname{tr}(\Sigma_R)^2}$. Here again, it is easy to see that the output of AltMin is significantly more accurate than the OLS solution. $\Sigma_R$ also plays a critical role here. In fact, if $\Sigma_R$ is nearly orthogonal to $\Sigma_*^{-1}$, then the gain over OLS is negligible. To understand this better, consider the following 2-task example:

$$y_i^1 = \langle \mathbf{x}_i, \mathbf{w}_* \rangle + \eta_i, \ \ y_i^2 = \langle \mathbf{x}_i, \mathbf{w}_* \rangle + \eta_i.$$

Note that the noise $\eta_i$ is perfectly correlated here. However, as rows $X_i^j = \mathbf{x}_i$ are also completely correlated. So, the two equations are just duplicates of each other and hence, AltMin cannot obtain any gains over OLS (as predicted by our bounds as well).

# 3 Seemingly Unrelated Regression

Seemingly-unrelated regression (SUR) model [21, 22] is a generalization of the basic linear regression model to handle vector valued outputs and has applications in several domains including multi-task learning, economics, genomics etc. Below we present the SUR model and our main result for estimating the coefficients in such a model.

Let $X_i \in \mathbb{R}^{m \times d}$, $1 \leq i \leq n$ be sampled i.i.d. from a fixed distribution $\mathcal{P}_X$. Let $W_* \in \mathbb{R}^{m \times d}$ be a fixed matrix of coefficients. The vector-valued output for each data point $X_i$ is given by:

$$\mathbf{y}_i = X_i \bullet W_* + \boldsymbol{\eta}_i, \tag{5}$$

where $X_i \bullet W_* = \operatorname{diag}(X_i W_*^T)$ and $\boldsymbol{\eta}_i \sim \mathcal{N}(0, \Sigma_*)$ is the noise vector with covariance $\Sigma_*$.

OLS and MLE solution can be defined similar to the Pooled model:

$$W_{OLS} = \arg \min_W \frac{1}{n} \sum_{i=1}^n \|\mathbf{y}_i - X_i \bullet W\|_2^2, \ \ W_{MLE} = \arg \min_W \frac{1}{n} \sum_{i=1}^n \left\| \Sigma_*^{-\frac{1}{2}} (\mathbf{y}_i - X_i \bullet W) \right\|_2^2. \tag{6}$$

Here again, we expect MLE to provide significantly better estimation of $W_*$ by exploiting noise correlation. As $\Sigma_*$ is not available apriori, both $\Sigma_*$ and $W_*$ are estimated by solving the following MLE problem:

$$(\widehat{W}, \widehat{\Sigma}) = \arg \max_{W, \Sigma} - \log |\Sigma| - \frac{1}{n} \sum_{i=1}^n \left\| \Sigma^{-\frac{1}{2}} (\mathbf{y}_i - X_i \bullet W) \right\|_2^2 \tag{7}$$

Here again, the MLE problem is non-convex and hence standard analysis does not provide strong convergence guarantees. Still, alternating minimization (of negative log-likelihood) for $\widehat{W}, \widehat{\Sigma}$ leads

---
**Algorithm 2** AltMin-SUR: Alternating Minimization for SUR
---
**Require:** $\mathcal{D} = \{(X_1, \mathbf{y}_1) \ldots (X_{2nT}, \mathbf{y}_{2nT})\}$, Number of iterations: $T$
 1: Randomly partition $\mathcal{D} = \{\mathcal{D}_0^\Sigma, \mathcal{D}_0^W, \mathcal{D}_1^\Sigma, \mathcal{D}_1^W, \ldots, \mathcal{D}_T^\Sigma, \mathcal{D}_T^W\}$, where $|\mathcal{D}_t^{\mathbf{w}}| = |\mathcal{D}_t^\Sigma| = n, \forall t$
 2: Initialize $W_0 = 0$
 3: **for** $t = 0, \ldots, T - 1$ **do**
 4:    Covariance Estimation: $\widehat{\Sigma}_t = \frac{1}{n} \sum_{i \in \mathcal{D}_t^\Sigma} (\mathbf{y}_i - X_i \bullet W)(\mathbf{y}_i - X_i \bullet W)^T$
 5:    Least-squares Solution: $W_{t+1} = \arg\min_W \frac{1}{n} \sum_{i \in \mathcal{D}_t^W} \|\widehat{\Sigma}_t^{-\frac{1}{2}} (\mathbf{y}_i - X_i \bullet W)\|_2^2$
 6: **end for**
 7: **Output**: $W_T$
---

to accurate answers in practice. Below, we analyze the AltMin procedure (see Algorithm 2) and show that the finite sample error bound of AltMin matches (up to logarithmic factors) the error rate of the MLE solution. Similar to the previous section, we modify the standard AltMin procedure to include fresh samples at each step of the algorithm.

**Theorem 7** (Result for SUR Model). *Let $X_i \overset{i.i.d.}{\sim} \mathcal{P}_X, 1 \le i \le n$, where $\|X\|_{\psi_2}$ is the sub-Gaussian norm of each $X_i$. Let $\boldsymbol{\eta}_i \sim \mathcal{N}(0, \Sigma_*), \Sigma_* \succ 0$, and $W_* \in \mathbb{R}^{m \times d}$ be a fixed coefficients matrix. Let $W_T$ be the $T$-th iterate of Algorithm 2. Also, let $n \ge C \cdot md\|X\|_{\psi_2}$, where $C > 0$ is a global constant. Then, the following holds (w.p. $\ge 1 - T/n^{10}$):*

$$\mathbb{E}_{X \sim \mathcal{P}_X} \left[ \|\Sigma_*^{-\frac{1}{2}} (X \bullet W_T - X \bullet W_*)\|_2^2 \right] \le \frac{4C^2 d \log(n)}{n} \cdot m + \|\Sigma_*\|_2^2 \cdot 2^{-T}.$$

*Moreover, if $\mathcal{P}_X$ is such that each row $X^p$ is sampled **independently** and has zero mean, i.e., $X^p \perp X^q, \forall p, q$ and $\mathbb{E}_{X \sim \mathcal{P}_X}[X] = 0$, then the following holds (w.p. $\ge 1 - T/n^{10}$):*

$$\sum_{j=1}^m (\Sigma_*^{-1})_{jj} \mathbb{E}_{X^j \sim \mathcal{P}_X^j} \left[ \left\langle X^j, W_T^j - W_*^j \right\rangle^2 \right] \le \frac{4C^2 d \log(n)}{n} \cdot m + \|\Sigma_*\|_2^2 \cdot 2^{-T}.$$

**Remarks**: It is easy to obtain error bounds for OLS in this case as it solves each equation independently. In particular, standard single-output linear regression analysis [23] gives:

$$\mathbb{E}_X [\sum_j \frac{1}{(\Sigma_*)_{jj}} \langle W_{OLS}^j - W_*^j, X^j \rangle^2] \le \frac{Cd}{n} \cdot m. \tag{8}$$

The weight for each individual error term $\left\langle W_T^j - W_*^j, X^j \right\rangle^2$ in the AltMin error bound is $(\Sigma_*^{-1})_{jj}$ while it is $\frac{1}{(\Sigma_*)_{jj}}$ for OLS. Using Claim 20, $(\Sigma_*^{-1})_{jj} \ge \frac{1}{(\Sigma_*)_{jj}}$. Hence, the error terms of AltMin should be significantly smaller than that of OLS. Similar to Section 1, we now provide an illustrative example to demonstrate prediction accuracy of AltMin (and MLE) solution vs. OLS solution.

**Illustrative Example**: Consider a two-valued output SUR problem, where $X^1 \sim \mathcal{N}(0, I_{d \times d})$ and $X^2 \sim \mathcal{N}(0, I_{d \times d})$ are sampled independently and the noise covariance by:

$$\Sigma_* = \begin{bmatrix} 1 & 1 - \epsilon \\ 1 - \epsilon & 1 \end{bmatrix}, \Sigma_*^{-1} = \frac{1}{2\epsilon} \begin{bmatrix} 1 & -(1 - \epsilon) \\ -(1 - \epsilon) & 1 \end{bmatrix},$$

where $0 < \epsilon < 1$. We sample $n \ge 2d$ points from this model. That is, $\mathbf{y}_i = X_i \bullet W_* + \boldsymbol{\eta}_i, 1 \le i \le n$. Hence, the estimation error of AltMin and OLS are given by:

$$\|W_T - W_*\|_F^2 \le \frac{4Cd \log n}{n} \cdot \epsilon, \quad \|W_{OLS} - W_*\|_F^2 \le \frac{2Cd}{n}.$$

Clearly, the error of AltMin decreases to 0 as $\epsilon \to 0$ (and $n \ge Cd \log d$), i.e., as noise is getting more correlated. In contrast, the error bound of OLS is independent of $\epsilon$ and remains a large constant even for $\epsilon = 0$ and $n = O(d)$.

**Multivariate Regression Model**: We now briefly discuss the popular Multivariate Regression (MR) model (arises in many applications including multitask learning with shared regressors), where each output $\mathbf{y}_i \in \mathbb{R}^m$ is modeled as:

$$\mathbf{y}_i = W_* \mathbf{x}_i + \boldsymbol{\eta}_i, \tag{9}$$

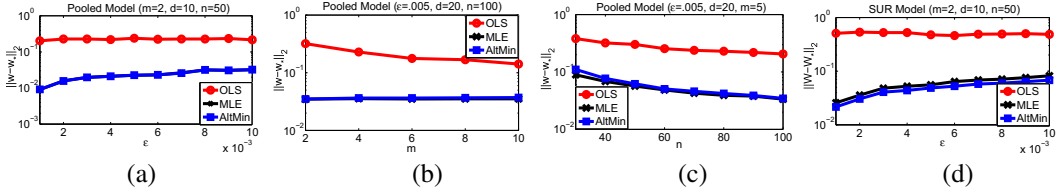

Figure 1: (a), (b), (c): Pooled Model. Estimation error $\|\mathbf{w} - \mathbf{w}_*\|_2$ for different algorithms (MLE, OLS, AltMin) with varying noise dependencies ($\epsilon$), $m$, $n$, (d): SUR Model. Comparison of estimation error $\|W - W_*\|_2$ with increasing $\epsilon$. Low $\epsilon$ implies badly conditioned noise covariance, i.e., $\text{tr}(\Sigma_*) \text{tr}(\Sigma_*^{-1}) \gg m^2$. In (a), (b), MLE and AltMin have almost overlapping error curves.

where $\mathbf{x}_i \in \mathbb{R}^d$ are all sampled i.i.d. from a fixed distribution and $\boldsymbol{\eta}_i \sim \mathcal{N}(0, \Sigma_*)$ is the noise vector. $W_* \in \mathbb{R}^{m \times d}$ is the coefficients matrix. Model (9) can be easily re-written as a SUR problem (5) where each row of $X_i$ is given by $\mathbf{x}_i^T$. That is, $X_i = [\mathbf{x}_i \mid \mathbf{x}_i \mid \ldots \mid \mathbf{x}_i]^T$. However, for the MR model, it is well known that the optimal solution to MLE problem is same as the OLS solution [6, Chapter 7]. Naturally, our AltMin/MLE error bounds also do not provide an advantage over OLS bounds. For *general* $W_*$ in the MR model, the MLE solution is independent of $\Sigma_*$. But, by imposing certain special structures on $W_*$, MLE indeed leads to significantly more accurate solution. For example, the Pooled and the SUR model can be posed as special cases of the MR model but with specially structured $W_*$. Similarly, other structures like reduced rank regression [24, 25] also allows exploitation of the noise correlation. We leave further investigation of other type of structural assumptions on $W_*$ as a topic of future research.

## 4 Experiments

In this section, we present results from simulations which were conducted with the following two-fold objective: a) demonstrate that both MLE and AltMin estimators indeed perform significantly better than the Ordinary Least Squares (OLS) estimator when the noise vector has significant dependencies, b) study scaling behavior of the three estimators (OLS, MLE, AltMin) w.r.t. $m, n$.

Solving MLE for the Pooled as well as SUR model is difficult in general. So, we we set $\Sigma = \Sigma_*$ in the MLE optimization ((4) for Pooled, (7) for SUR), where $\Sigma_*$ is the true noise covariance matrix. In this case, the estimator reduces to a least squares problem. We implemented all the three estimators in Matlab and provide results averaged over 20 runs. We run AltMin for at most 50 iterations.

*Pooled Model*: For the first set of experiments, we generated the data $(X_i, \mathbf{y}_i)$ using the Pooled Model (Section 2). We generated $X_i$'s from spherical multi-variate Gaussian and selected $\mathbf{w}_*$ to be a random vector. In the first sub-experiment, we considered $m = 2$ and set $\Sigma_* = \begin{bmatrix} 1 & 1 - \epsilon \\ 1 - \epsilon & 1 \end{bmatrix}$. Using Theorem 2, AltMin as well as MLE estimator should have error $\|\mathbf{w} - \mathbf{w}_*\|_2^2 \leq \frac{Cd}{n} \cdot (\epsilon - \epsilon^2)$ while for OLS it is $\|\mathbf{w} - \mathbf{w}_*\|_2^2 \leq \frac{Cd}{n}$. Figure 4 (a) shows that our simulations also exhibit exactly the same trend as predicted by our error bounds. Moreover, errors of both MLE and AltMin are exactly the same, indicating that AltMin indeed converged to the MLE estimate.

Next, we set $\Sigma_*$ as:
$$\Sigma_* = \begin{bmatrix} 1 & 1 - \epsilon & \mathbf{0} \\ 1 - \epsilon & 1 & \mathbf{0} \\ \mathbf{0} & \mathbf{0} & I_{m-2 \times m-2} \end{bmatrix}, \tag{10}$$

with $\epsilon = 0.005$ and measure recovery error ($\|\mathbf{w} - \mathbf{w}_*\|_2$) while varying $m$ and $n$. Note that for AltMin and MLE, the error bound for such $\Sigma_*$ is $\|\mathbf{w} - \mathbf{w}_*\|_2^2 \leq \frac{Cd}{n} \cdot \frac{(\epsilon - \epsilon^2)}{(m-2)(\epsilon - \epsilon^2) + 1}$ and hence AltMin and MLE's error does not change significantly with increasing $m$. But for OLS the error goes down with $m$ as $\|\mathbf{w} - \mathbf{w}_*\|_2^2 \leq \frac{Cd}{n} \cdot \frac{1}{m}$ which can be observed in the Figure 4(b) as well. Finally, Figure 4(c) clearly indicates that $\|\mathbf{w} - \mathbf{w}_*\| = O(\frac{1}{\sqrt{n}})$ for all the three methods, hence matching our theoretical bounds.

*SUR Model*: Here we generated data $(X_i, \mathbf{y}_i)$ using the SUR model (Section 3) but with $X_i$ sampled from spherical Gaussians. $W_*$ was selected to be a random Gaussian matrix. $\Sigma_*$ is given by (10). As illustrated in Section 3, the error of MLE/AltMin is at most $O(\epsilon)$ while the error of OLS is independent of $\epsilon$. Figure 4 (d) clearly demonstrates the above mentioned error trends.

## Footnotes

[1]For simplicity of exposition, throughout the remaining paper, we assume that $\Sigma_*$ is invertible. Non-invertible $\Sigma_*$ can be handled using simple limit arguments and in fact, our results get significantly better if $\Sigma_*$ is not invertible

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
