[Supplementary Material]

# A High-dimensional Setting

We now study the vector-output regression problems in the high-dimensional setting, where $d \gg n$ but the parameter vector is required to be $s$-sparse with $s \ll d$. Our goal is to provide an algorithm with error bounds that are at most logarithmic in $d$ and are linear in $s$. Here, we provide our result for a special case of the Pooled model, where data is sampled from a Gaussian distribution. Our analysis can be easily extended to the SUR model as well.

Let $X_i = \Sigma_R^{\frac{1}{2}} Z_i \Lambda^{\frac{1}{2}}$ where each entry of $Z_i$ is sampled i.i.d. from the univariate normal distribution and $\Sigma_R \succ 0$, $\Lambda \succ 0$. Let $\mathbf{w}_* \in \mathbb{R}^d$ be such that $\mathbf{w}_*$ is $s$-sparse, i.e., $\|\mathbf{w}_*\|_0 \leq s$. The outputs are given by, $\mathbf{y}_i = X_i \mathbf{w}_* + \boldsymbol{\eta}_i$, $\boldsymbol{\eta}_i \sim \mathcal{N}(0, \Sigma_*)$, where $\Sigma_* \succ 0$.

For the above setting, we analyze Algorithm 1, but where Least Squares Estimation step (Step 5) is replaced by sparsity constrained optimization:

$$\widehat{\mathbf{w}} = \arg \min_{\|\mathbf{w}\|_0 \leq s} f(\mathbf{w}) = \arg \min_{\|\mathbf{w}\|_0 \leq s} \frac{1}{n} \sum_{i \in \mathcal{D}_t^{\mathbf{w}}} \|\widehat{\Sigma}_t^{-\frac{1}{2}}(\mathbf{y}_i - X_i \mathbf{w})\|_2^2 \qquad (11)$$

Note that the above problem is in general NP-hard to solve due to the sparsity constraint. But, we can use the Iterative Hard Thresholding (IHT) method [26] to solve (11), if $f(w)$ satisfies the restricted strong smoothness (RSS) and the restricted strong convexity (RSC) properties (defined in (12)). Below, we re-state the IHT convergence result by by [26].

**Theorem 8** (Theorem 1 of [26]). *Let $f$ have RSC and RSS parameters given by $L_{3\widetilde{s}} = L$ and $\alpha_{3\widetilde{s}} = \alpha$ respectively. Let IHT algorithm (Algorithm 1, [26]) be invoked with $f$, $\widetilde{s} = \kappa^2 \cdot s$. Then, the $\tau$-th iterate of IHT ($\mathbf{w}_{t+1}$), for $\tau = O(\frac{L}{\alpha} \cdot \log \frac{\|\Sigma_*\|_2}{\epsilon})$ satisfies: $f(\mathbf{w}_{t+1}) \leq f(\widehat{\mathbf{w}}) + \epsilon$, where $\widehat{\mathbf{w}}$ is any global optimum of* (11).

As the algorithm has only logarithmic dependence on $\epsilon$, we can set $\epsilon$ to be arbitrary small (say $.001 f(\widehat{\mathbf{w}})$). For simplicity, we ignore $\epsilon$ for now. Note that the proof of Lemma 3 only requires that the least squares step satisfies: $f(\mathbf{w}_{t+1}) \leq f(\mathbf{w}_*)$. Moreover, columns of $X$ corresponding to the index set $\mathcal{S}_{t+1} = \text{supp}(\mathbf{w}_t) \cup \text{supp}(\mathbf{w}_{t+1}) \cup \text{supp}(\mathbf{w}_*)$ are used by $\widehat{\Sigma}_t$ and the least squares solution. So, Lemma 3 applies directly but with $d = |\mathcal{S}_{t+1}| \leq 3\widetilde{s}^2$.

Hence, we obtain the following error bound for the $T$-th iterate of Algorithm 1:

$$\mathbb{E}_{X \sim \mathcal{P}_X}[\|X(\mathbf{w}_T - \mathbf{w}_*)\|_2^2] \leq \frac{8C\widetilde{s} \log d}{n} \cdot \frac{m}{\text{tr}(\Sigma_R \Sigma_*^{-1})} + 2^{-T},$$

Recall that $\widetilde{s} = \left(\frac{L}{\alpha}\right)^2 \cdot s$, where $L, \alpha$ are the RSS and the RSC constants of $f$. Hence, we now only need to provide RSS/RSC constants for the above given $f$.

**Lemma 9** (RSC/RSS). *Let $X_i$ be as given above. Also, let $n \geq C\widehat{s} \log d$. Then the following holds for all fixed $A$ (w.p. $\geq 1 - \exp(-n)$):*

$$0.5 \cdot \lambda_{\min}(\Lambda) \text{tr}(A^T A \Sigma_R) \|\mathbf{v}\|_2^2 \leq \frac{1}{n} \sum_{i=1}^n \mathbf{v}^T X_i^T A^T A X_i \mathbf{v} \leq 2 \text{tr}(A^T A \Sigma_R) \cdot \lambda_{\max}(\Lambda) \|\mathbf{v}\|_2^2,$$

*where $\mathbf{v} \in \mathbb{R}^d$ is **any** $\widehat{s}$-sparse vector.*

The above lemma shows that for any $\widehat{s}$-sparse $\mathbf{w}, \mathbf{w}'$, we have:

$$\frac{\alpha_{2\widehat{s}}}{2} \|\mathbf{w} - \mathbf{w}'\|_2^2 + \langle \nabla f, \mathbf{w} - \mathbf{w}' \rangle + f(\mathbf{w}') \leq f(\mathbf{w}) \leq f(\mathbf{w}') + \langle \nabla f, \mathbf{w} - \mathbf{w}' \rangle + \frac{L_{2\widehat{s}}}{2} \|\mathbf{w} - \mathbf{w}'\|_2^2, \quad (12)$$

where $L = L_{2\widehat{s}} = 2\lambda_{\max}(\Lambda)$ is the RSS constant of $f$ and $\alpha = \alpha_{2\widehat{s}} = \frac{\lambda_{\min}(\Lambda)}{2}$ is the RSC constant.

That is the error bound for AltMin procedure is given by:

$$\mathbb{E}_{X \sim \mathcal{P}_X}[\|X(\mathbf{w}_T - \mathbf{w}_*)\|_2^2] \leq \frac{8Cs \log d}{n} \cdot \left(\frac{\lambda_{\max}(\Lambda)}{\lambda_{\min}(\Lambda)}\right)^2 \cdot \frac{m}{\text{tr}(\Sigma_R \Sigma_*^{-1})} + 2^{-T}. \qquad (13)$$

Note that the above bound is linear in $s$ but has a condition number (of $\Lambda$) dependence. The condition number factor appears in the analysis of the standard linear regression as well [26], and is in general unavoidable for computationally efficient algorithms [27].

*Proof of Lemma 9.* Consider a "fixed" support set $\mathcal{S}$ s.t. $|\mathcal{S}| = \widehat{s}$. Also, let $\Sigma_R^{1/2} A^T A \Sigma_R^{1/2} = \sum_j \widehat{\lambda}_j \mathbf{u}_j \mathbf{u}_j^T$ be the eigenvalue decomposition of $A^T A$. Then, w.p. $\geq 1 - \exp(-n)$, the following holds for all $\mathbf{v} \in \mathbb{R}^d$ s.t. $\|\mathbf{v}\|_0 \leq \widehat{s}$ and $\mathrm{supp}(\mathbf{v}) \subseteq \mathcal{S}$:

$$
\begin{aligned}
\mathbf{v}^T \sum_{i=1}^n X_i A^T A X_i \mathbf{v} &= \sum_{j=1}^m \widehat{\lambda}_j \mathbf{v}_{\mathcal{S}}^T \left( \sum_{i=1}^n (X_i^T)_{\mathcal{S}} \mathbf{u}_j \mathbf{u}_j^T (X_i)_{\mathcal{S}} \right) \mathbf{v}_{\mathcal{S}}, \\
&= \sum_{j=1}^m \widehat{\lambda}_j \mathbf{v}_{\mathcal{S}}^T \Lambda_{\mathcal{S}\mathcal{S}}^{\frac{1}{2}} \left( \sum_{i=1}^n \Lambda_{\mathcal{S}\mathcal{S}}^{-\frac{1}{2}} (X_i)_{\mathcal{S}}^T \mathbf{u}_j \mathbf{u}_j^T (X_i)_{\mathcal{S}} \Lambda_{\mathcal{S}\mathcal{S}}^{-\frac{1}{2}} \right) \Lambda_{\mathcal{S}\mathcal{S}}^{\frac{1}{2}} \mathbf{v}_{\mathcal{S}}, \\
&\overset{\zeta_1}{\geq} \frac{n}{2} \sum_{j=1}^m \widehat{\lambda}_j \mathbf{v}_{\mathcal{S}}^T \Lambda_{\mathcal{S}\mathcal{S}} \mathbf{v}_{\mathcal{S}} = \frac{n}{2} \mathrm{tr}(A^T A \Sigma_R) \mathbf{v}^T \Lambda \mathbf{v}, \quad (14)
\end{aligned}
$$

where $\zeta_1$ follows by Lemma 10, $(X_i)_{\mathcal{S}}$ denotes the submatrix of $X_i$ corresponding to index set $\mathcal{S}$. Similarly, $\mathbf{v}_{\mathcal{S}}$ and $\Lambda_{\mathcal{S}\mathcal{S}}$ can be defined to be sub-vector and sub-matrix of $\mathbf{v}$ and $\Lambda$, respectively. The lower bound of the theorem follows by taking a union bound on all $O(d^{\widehat{s}})$ sets $\mathcal{S}$ and setting $n \geq C \widehat{s} \log d$.

The upper bound on $\mathbf{v}^T \left( \frac{1}{n} \sum_{i=1}^n X_i^T A^T A X_i \right) \mathbf{v}$ also follows similarly. $\quad\square$

## B  Technical Lemmas

**Lemma 10.** *Let $\mathbf{z}_i \overset{i.i.d.}{\sim} \mathcal{P}_{\mathbf{z}}, 1 \leq i \leq n$, where $\mathcal{P}_{\mathbf{z}}$ is such that $\mathbb{E}_{\mathbf{z} \sim \mathcal{P}_{\mathbf{z}}}[\mathbf{z}\mathbf{z}^T] = I_{d \times d}$ and the sub-Gaussian norm of $\mathbf{z}$ is given by $\|\mathbf{z}\|_{\psi_2}$. Let $n \geq Cd \cdot \|\mathbf{z}\|_{\psi_2}$. Then, the following holds w.p. $\geq 1 - \exp(-C \cdot n)$:*

$$
\left\| \frac{1}{n} \sum_{i=1}^n \mathbf{z}_i \mathbf{z}_i^T - I_{d \times d} \right\|_2 \leq \frac{1}{10}.
$$

*Proof.* Lemma follows directly by Corollary 5.50 of [28]. $\quad\square$

**Lemma 11** (Corollary 5.35 of [28])**.** *Let $M \in \mathbb{R}^{m \times n}$ be s.t. $M_{ij} \overset{i.i.d.}{\sim} \mathcal{N}(0,1), \forall i, j$. Also, let $n \geq 4m$, then the following holds w.p. $\geq 1 - \exp(-C \cdot n)$:*

$$
\frac{1}{2}\sqrt{n} \leq \sigma_m(M) \leq \sigma_1(M) \leq 2\sqrt{n},
$$

*where $\sigma_i(M)$ is the $i$-th singular value of $M$.*

**Lemma 12.** *Let $\mathbf{g} = A\boldsymbol{\eta}$, where $\boldsymbol{\eta} \sim \mathcal{N}(0, I_{n \times n}) \in \mathbb{R}^n$ and $A \in \mathbb{R}^{m \times n}$ is a fixed matrix independent of $\boldsymbol{\eta}$. Also, let $n \geq Cm$. Then, w.p. $\geq 1 - 1/n^{10}$, we have:*

$$
\|\mathbf{g}\|_2^2 \leq C\|A\|_F^2 \log(n).
$$

*Proof.* First consider the $j$-th coordinate of $\mathbf{g}_j = \mathbf{e}_j^T A\boldsymbol{\eta}$. As $\boldsymbol{\eta}$ is a Gaussian vector and $A$ is a fixed matrix, $\mathbf{g}_j \sim \|\mathbf{e}_j^T A\|_2 \cdot \mathcal{N}(0,1)$. Hence, w.p. $1 - 1/n^{11}$, $\mathbf{g}_j \leq C\|\mathbf{e}_j^T A\|_2 \sqrt{\log n}$. Lemma now follows by combining the above observation with $\|\mathbf{g}\|_2^2 = \sum_j (\mathbf{e}_j^T A\boldsymbol{\eta})^2$ and the union bound. $\quad\square$

**Lemma 13.** *Let $\mathbf{g} = \sum_i A_i \boldsymbol{\eta}_i$, where $A_i \in \mathbb{R}^{d \times m}$, $\boldsymbol{\eta}_i \overset{i.i.d.}{\sim} \mathcal{N}(0, I_{m \times m}) \in \mathbb{R}^m$. Also, let $n \geq Cm$. Then, w.p. $\geq 1 - 1/n^{10}$, we have:*

$$
\|\mathbf{g}\|_2^2 \leq C \left( \sum_{i=1}^n \|A_i\|_F^2 \right) \log n.
$$

*Proof.* Lemma now follows by applying Lemma 12 to $\mathbf{g} = A\boldsymbol{\eta}$ where $A = [A_1 \ \ldots \ A_n] \in \mathbb{R}^{d \times mn}$, $\boldsymbol{\eta} = [\boldsymbol{\eta}_1^T \ \cdots \ \boldsymbol{\eta}_n^T]^T$. $\qquad\qquad\qquad\qquad\qquad\qquad\qquad\qquad\qquad\qquad\qquad\quad$ $\square$

**Lemma 14.** *Let $X_i \sim \mathcal{P}_X, 1 \leq i \leq n$ be sub-Gaussian random variables. Also, let $n \geq Cd\|X\|_{\psi_2} \log d$, where $\|X\|_{\psi_2}$ is the sub-Gaussian norm of each $X_i$ (see Definition 1). Let $A$ be any **fixed** matrix. Then, w.p. $\geq 1 - m\exp(-C \cdot n)$, the following holds **for all $\mathbf{v} \in \mathbb{R}^d$**:*

$$\frac{1}{2}\mathbf{v}^T \left(E_{X\sim\mathcal{P}_X}\left[X^T A^T A X\right]\right)\mathbf{v} \leq \mathbf{v}^T \left(\frac{1}{n}\sum_{i=1}^n X_i^T A^T A X_i\right)\mathbf{v} \leq 2\mathbf{v}^T \left(E_{X\sim\mathcal{P}_X}\left[X^T A^T A X\right]\right)\mathbf{v}.$$

*Proof.* Let $A^T A = \sum_j \lambda_j(A^T A)\mathbf{u}_j \mathbf{u}_j^T$ be the eigenvalue decomposition of $A^T A$. Then, $\forall \mathbf{v} \in \mathbb{R}^d$:

$$\mathbf{v}^T \sum_{i=1}^n X_i A^T A X_i \mathbf{v} = \sum_{j=1}^m \lambda_j(A^T A)\mathbf{v}^T \left(\sum_{i=1}^n X_i^T \mathbf{u}_j \mathbf{u}_j^T X_i\right)\mathbf{v}$$

$$= \sum_{j=1}^m \lambda_j(A^T A)\mathbf{v}^T \Sigma_{Xu_j}^{\frac{1}{2}} \left(\sum_{i=1}^n \Sigma_{Xu_j}^{-\frac{1}{2}} X_i^T \mathbf{u}_j \mathbf{u}_j^T X_i \Sigma_{Xu_j}^{-\frac{1}{2}}\right)\Sigma_{Xu_j}^{\frac{1}{2}}\mathbf{v}, \quad (15)$$

where $\Sigma_{X\mathbf{u}_j} = \mathbb{E}_{X\sim\mathcal{P}_X}[X^T \mathbf{u}_j \mathbf{u}_j^T X]$. Let $\mathbf{z}_{ij} = \Sigma_{Xu_j}^{-\frac{1}{2}} X_i^T \mathbf{u}_j$. Then, by definition of $\Sigma_{X\mathbf{u}_j}$, we have:

$$\mathbb{E}[\mathbf{z}_{ij}\mathbf{z}_{ij}^T] = I_{d\times d}.$$

Moreover, $\|\mathbf{z}_{ij}\|_{\psi_2} \leq \|X\|_{\psi_2}$ by definition (see Definition 1). Hence, using Lemma 10 and the union bound for $m$ $\mathbf{u}_j$'s (recall that $A$ and hence $\mathbf{u}_j$'s are fixed), w.p. $\geq 1 - m\exp(-Cn)$ the following holds for all $\mathbf{v} \in \mathbb{R}^d$:

$$\mathbf{v}^T \sum_{i=1}^n X_i A^T A X_i \mathbf{v} \geq \frac{n}{2}\sum_{j=1}^m \lambda_j(A^T A)\mathbf{v}^T \Sigma_{Xu_j}\mathbf{v} = \frac{n}{2}\sum_{j=1}^m \lambda_j(A^T A)\mathbf{v}^T \mathbb{E}_{X\sim\mathcal{P}_X}[X\mathbf{u}_j \mathbf{u}_j^T X]\mathbf{v},$$

$$= \frac{n}{2}\mathbf{v}^T \left(\mathbb{E}_{X\sim\mathcal{P}_X}[X^T A^T A X]\right)\mathbf{v}. \quad (16)$$

The upper bound on $\mathbf{v}^T \left(\frac{1}{n}\sum_{i=1}^n X_i^T A^T A X_i\right)\mathbf{v}$ also follows similarly. $\qquad\qquad\qquad$ $\square$

## C Proofs of Claims from Section 2

We first provide analysis for a general estimator that decorrelates noise using certain *fixed $A, B$* matrices. Our bounds for OLS, MLE follow directly using the below given general theorem.

**Theorem 15.** *Let $X_i \overset{i.i.d.}{\sim} \mathcal{P}_X, 1 \leq i \leq n$ where $\mathcal{P}_X$ is a sub-Gaussian distribution with sub-Gaussian norm $\|X\|_{\psi_2} < \infty$ (see Definition 1). Also, let $\boldsymbol{\eta}_i \sim \mathcal{N}(0, I_{m\times m})$. Let $\mathbf{w}_* \in \mathbb{R}^d$ be a fixed weight vector and $A, B$ be fixed matrices. Let,*

$$\widehat{\mathbf{w}} = \arg\min_{\mathbf{w}} \frac{1}{n}\sum_{i=1}^n \|AX_i(\mathbf{w} - \mathbf{w}_*) - B\boldsymbol{\eta}_i\|_2^2. \quad (17)$$

*Also, let $n \geq C \cdot (m+d)\log(m+d) \cdot \|X\|_{\psi_2}$, where $C > 0$ is a global constant. Then, the following holds (w.p. $\geq 1 - 1/n^{10}$):*

$$\mathbb{E}_{X\sim\mathcal{P}_X}\left[\|AX(\widehat{\mathbf{w}} - \mathbf{w}_*)\|_2^2\right] \leq \frac{C^2 d\log(n)}{n} \cdot \|B\|_2^2.$$

*Proof.* As $\widehat{\mathbf{w}}$ is the optimal solution to (17), we have:

$$\sum_{i=1}^{n} \|AX_i(\widehat{\mathbf{w}} - \mathbf{w}_*) - B\boldsymbol{\eta}_i\|_2^2 \leq \sum_{i=1}^{n} \|B\boldsymbol{\eta}_i\|_2^2,$$

$$\sum_{i=1}^{n} \|AX_i(\widehat{\mathbf{w}} - \mathbf{w}_*)\|_2^2 \leq 2(\widehat{\mathbf{w}} - \mathbf{w}_*)^T F^{\frac{1}{2}} F^{-\frac{1}{2}} \sum_{i=1}^{n} X_i^T A^T B\boldsymbol{\eta}_i,$$

$$\|F^{\frac{1}{2}}(\widehat{\mathbf{w}} - \mathbf{w}_*)\|_2^2 \overset{\zeta_1}{\leq} 2\|F^{\frac{1}{2}}(\widehat{\mathbf{w}} - \mathbf{w}_*)\|_2 \|F^{-\frac{1}{2}} \sum_{i=1}^{n} X_i^T A^T B\boldsymbol{\eta}_i\|_2, \qquad (18)$$

where $F = \sum_{i=1}^{n} X_i^T A^T A X_i$ and $\zeta_1$ follows from Cauchy-Schwarz inequality. Also, using Lemma 14, $\lambda_{\min}(F) \geq \frac{1}{2}\lambda_{\min}(\mathbb{E}_{X \sim \mathcal{P}_X}[X^T A^T A X])$. Using the fact that $\|X\|_{\psi_2} < \infty$ (see Definition 1), we have $\lambda_{\min}(F) \geq \frac{1}{2}\lambda_{\min}(\mathbb{E}_{X \sim \mathcal{P}_X}[X^T A^T A X]) > 0$. Hence, $F^{-1/2}$ is well-defined.

Note that $\mathbf{g} = \sum_{i=1}^{n} F^{-\frac{1}{2}} X_i^T A^T B\boldsymbol{\eta}_i$. Using Lemma 13, we have (w.p. $\geq 1 - 1/n^{10}$):

$$\|\mathbf{g}\|_2^2 \leq \log n \cdot \sum_{i=1}^{n} \|F^{-\frac{1}{2}} X_i^T A^T B\|_F^2 = \log n \cdot \mathrm{tr}\left(\sum_{i=1}^{n} X_i F^{-1} X_i^T A^T B B^T A\right),$$

$$\leq \log n \cdot \|B\|_2^2 \, \mathrm{tr}\left(F^{-1} \sum_{i=1}^{n} X_i^T A^T A X_i\right),$$

$$= d \log n \cdot \|B\|_2^2, \qquad (19)$$

where the last equality follows from the definition of $F$.

Now, using Lemma 14, we have (w.p. $\geq 1 - m \exp(-Cn)$):

$$(\widehat{\mathbf{w}} - \mathbf{w}_*)^T \sum_{i=1}^{n} X_i^T A^T A X_i(\widehat{\mathbf{w}} - \mathbf{w}_*) \geq \frac{n}{2}(\widehat{\mathbf{w}} - \mathbf{w}_*)^T \left(\mathbb{E}_{X \sim \mathcal{P}_X}[X^T A^T A X]\right)(\widehat{\mathbf{w}} - \mathbf{w}_*). \quad (20)$$

Theorem now follows by combining (18), (19), and (20). $\qquad\square$

We now provide proofs of both Theorem 2 as well as Lemma 3 which is the key component used by our proof of the main theorem.

*Proof of Theorem 2.* Theorem follows using Lemma 3 and observing that:

$$\mathbb{E}_{X \sim \mathcal{P}_X}\left[\|\Sigma_*^{-\frac{1}{2}} X(\mathbf{w}_T - \mathbf{w}_*)\|_2^2\right] = \mathbb{E}_{X \sim \mathcal{P}_X}\left[\|\Sigma_*^{-\frac{1}{2}} X \Sigma_X^{-\frac{1}{2}} \Sigma_X^{\frac{1}{2}}(\mathbf{w}_T - \mathbf{w}_*)\|_2^2\right]$$

$$\geq \lambda_{\min}\left(\Sigma_{X*}\right) \|\Sigma_X^{\frac{1}{2}}(\mathbf{w}_T - \mathbf{w}_*)\|_2^2,$$

$$= \lambda_{\min}\left(\Sigma_{X*}\right) \mathbb{E}_{X \sim \mathcal{P}_X}\left[\|X(\mathbf{w}_T - \mathbf{w}_*)\|_2^2\right], \qquad (21)$$

where the second inequality follows from the definition of $\Sigma_{X*}$ and the last equality follows by using $\Sigma_X = \mathbb{E}_{X \sim \mathcal{P}_X}[X^T X]$. $\qquad\square$

*Proof of Lemma 3.* Recall that,

$$\widehat{\Sigma}_t = \frac{1}{n} \sum_{i \in \mathcal{D}_t^\Sigma} (\mathbf{y}_i - X_i \mathbf{w}_t)(\mathbf{y}_i - X_i \mathbf{w}_t)^T, \quad \Sigma_t = \Delta + \Sigma_*,$$

where $\Delta = \frac{1}{n} \sum_{i \in \mathcal{D}_t^\Sigma} X_i(\mathbf{w}_* - \mathbf{w}_t)(\mathbf{w}_* - \mathbf{w}_t)^T X_i^T$. Now, using Lemma 14, $\lambda_{\min}(\widehat{\Sigma}_t) \geq \frac{1}{2}\lambda_{\min}(\Sigma_t) > 0$ as $\Sigma_* \succ 0$. So, $\widehat{\Sigma}_t$ is invertible.

Note that the samples of set $\mathcal{D}_t^\Sigma$ are independent of $\mathbf{w}_t$ as well as $\mathcal{D}_t^{\mathbf{w}}$. Moreover, $\mathcal{D}_t^{\mathbf{w}}$ is independent of $\mathbf{w}_t$ and hence is independent of $\Delta$. Also,

$$\mathbf{w}_{t+1} = \arg\min_{\mathbf{w}} \frac{1}{n} \sum_{i \in \mathcal{D}_t^{\mathbf{w}}} \|\widehat{\Sigma}_t^{-\frac{1}{2}} X_i(\mathbf{w} - \mathbf{w}_*) - \widehat{\Sigma}_t^{-\frac{1}{2}} \boldsymbol{\eta}_i\|_2^2. \qquad (22)$$

Let $A = \widehat{\Sigma}_t^{-\frac{1}{2}}$ and $B = \widehat{\Sigma}_t^{-\frac{1}{2}}\Sigma_*^{\frac{1}{2}}$. Note that, $A$, $B$ are both *fixed* matrices w.r.t. $\mathcal{D}_t^{\mathbf{w}}$ as $\Delta$ itself is independent of $\mathcal{D}_t^{\mathbf{w}}$. Now, applying Theorem 15 with the above mentioend $A$, $B$, we get (w.p. $\geq 1 - 1/n^{10}$):

$$\mathbb{E}_{X \sim \mathcal{P}_X}\left[\|\widehat{\Sigma}_t^{-\frac{1}{2}}X(\mathbf{w}_{t+1} - \mathbf{w}_*)\|_2^2\right] \leq \frac{C^2 d \log(n)}{n} \cdot \|\Sigma_*^{\frac{1}{2}}\widehat{\Sigma}_t^{-1}\Sigma_*^{\frac{1}{2}}\|_2. \tag{23}$$

Now, the following holds (w.p. $\geq 1 - \exp(-cn)$):

$$\|\Sigma_*^{\frac{1}{2}}\widehat{\Sigma}_t^{-1}\Sigma_*^{\frac{1}{2}}\|_2 = \max_{\mathbf{v}, \|\mathbf{v}\|_2 = 1} \mathbf{v}^T \Sigma_*^{\frac{1}{2}}\widehat{\Sigma}_t^{-1}\Sigma_*^{\frac{1}{2}}\mathbf{v} \overset{\zeta_1}{\leq} \max_{\mathbf{v}, \|\mathbf{v}\|_2 = 1} 2\mathbf{v}^T \Sigma_*^{\frac{1}{2}}\Sigma_t^{-1}\Sigma_*^{\frac{1}{2}}\mathbf{v},$$

$$= 2\|(I_{m \times m} + \Sigma_*^{-\frac{1}{2}}\Delta\Sigma_*^{-\frac{1}{2}})^{-1}\|_2 \leq 2, \tag{24}$$

where $\zeta_1$ follows from Lemma 16 and the fact that $\mathcal{D}_t^{\Sigma}$ is independent of $\mathbf{w}_t$. The last inequality follows as $\Delta$ is a p.s.d. matrix, so, $\lambda_{\min}(I_{m \times m} + \Sigma_*^{-\frac{1}{2}}\Delta\Sigma_*^{-\frac{1}{2}}) \geq 1$.

Next, we have:

$$\mathbb{E}_{X \sim \mathcal{P}_X}\left[\|\widehat{\Sigma}_t^{-\frac{1}{2}}X(\widehat{\mathbf{w}} - \mathbf{w}_*)\|_2^2\right] \overset{\zeta_1}{\geq} \frac{1}{2}\mathbb{E}_{X \sim \mathcal{P}_X}\left[\|\Sigma_t^{-\frac{1}{2}}X(\widehat{\mathbf{w}} - \mathbf{w}_*)\|_2^2\right],$$

$$= \frac{1}{2}\mathbb{E}_{X \sim \mathcal{P}_X}\left[\|\Sigma_t^{-\frac{1}{2}}\Sigma_*^{\frac{1}{2}}\Sigma_*^{-\frac{1}{2}}X(\widehat{\mathbf{w}} - \mathbf{w}_*)\|_2^2\right],$$

$$\overset{\zeta_2}{=} \frac{1}{2}\mathbb{E}_{X \sim \mathcal{P}_X}\left[(\widehat{\mathbf{w}} - \mathbf{w}_*)^T X^T \Sigma_*^{-\frac{1}{2}}\left(I_{m \times m} + \Sigma_*^{-\frac{1}{2}}\Delta\Sigma_*^{-\frac{1}{2}}\right)^{-1}\Sigma_*^{-\frac{1}{2}}X(\widehat{\mathbf{w}} - \mathbf{w}_*)\right],$$

$$\geq \frac{1}{\lambda_{\max}\left(I_{m \times m} + \Sigma_*^{-\frac{1}{2}}\Delta\Sigma_*^{-\frac{1}{2}}\right)}\mathbb{E}_{X \sim \mathcal{P}_X}\left[\|\Sigma_*^{-\frac{1}{2}}X(\widehat{\mathbf{w}} - \mathbf{w}_*)\|_2^2\right], \tag{25}$$

where $\zeta_1$ follows from Lemma 16 and $\zeta_2$ follows from the definition of $\Sigma_t$, and the last equation follows from $\lambda_{\min}\left((I_{m \times m} + \Sigma_*^{-\frac{1}{2}}\Delta\Sigma_*^{-\frac{1}{2}})^{-1}\right) = \frac{1}{\lambda_{\max}\left(I_{m \times m} + \Sigma_*^{-\frac{1}{2}}\Delta\Sigma_*^{-\frac{1}{2}}\right)}$.

Now,

$$\lambda_{\max}\left(I_{m \times m} + \Sigma_*^{-\frac{1}{2}}\Delta\Sigma_*^{-\frac{1}{2}}\right) = 1 + \|\Sigma_*^{-\frac{1}{2}}\Delta\Sigma_*^{-\frac{1}{2}}\|_2$$

$$\leq 1 + \frac{1}{n}\sum_{i=1}^{n} \text{tr}(\Sigma_*^{-\frac{1}{2}}X_i(\mathbf{w}_t - \mathbf{w}_*)(\mathbf{w}_t - \mathbf{w}_*)^T X_i^T \Sigma_*^{-\frac{1}{2}})$$

$$= 1 + (\mathbf{w}_t - \mathbf{w}_*)^T \left(\frac{1}{n}\sum_{i=1}^{n} X_i^T \Sigma_*^{-1} X_i\right)(\mathbf{w}_t - \mathbf{w}_*)$$

$$\overset{\zeta_1}{\leq} 1 + 2\mathbb{E}_{X \sim \mathcal{P}_X}\left[\|\Sigma_*^{-\frac{1}{2}}X(\mathbf{w}_t - \mathbf{w}_*)\|_2^2\right], \tag{26}$$

where $\zeta_1$ follows from Lemma 14.

Using (23), (24), (25), and (26), we have:

$$\mathbb{E}_{X \sim \mathcal{P}_X}\left[\|\Sigma_*^{-\frac{1}{2}}X(\mathbf{w}_{t+1} - \mathbf{w}_*)\|_2^2\right] \leq \frac{2C^2 d \log(n)}{n} + \frac{4C^2 d \log(n)}{n} \cdot \mathbb{E}_{X \sim \mathcal{P}_X}\left[\|\Sigma_*^{-\frac{1}{2}}X(\mathbf{w}_t - \mathbf{w}_*)\|_2^2\right].$$

Theorem now follows, as $n \geq 16Cd \log d$. $\qquad \square$

*Proof of Lemma 5.* We show the error bound for a general estimator:

$$\widehat{\mathbf{w}} = \arg\min_{\mathbf{w}} \frac{1}{n}\sum_{i=1}^{n} \|AX_i(\mathbf{w} - \mathbf{w}_*) - B\eta_i\|_2^2,$$

where $A, B \in \mathbb{R}^{m \times m}$ are fixed p.s.d. matrices.

Now, the optimal solution is given by:

$$\left(\frac{1}{n}\sum_{i=1}^{n} X_i^T A^T A X_i\right)(\widehat{\mathbf{w}} - \mathbf{w}_*) = \frac{1}{n}\sum_{i=1}^{n} X_i^T A^T B \eta_i,$$

$$\left(\frac{1}{n}\sum_{i=1}^{n} \Lambda^{-\frac{1}{2}} X_i^T A^T A X_i \Lambda^{-\frac{1}{2}}\right)\Lambda^{\frac{1}{2}}(\widehat{\mathbf{w}} - \mathbf{w}_*) = \frac{1}{n}\sum_{i=1}^{n} \Lambda^{-\frac{1}{2}} X_i^T A^T B \eta_i. \qquad (27)$$

Note that, $X_i = Z_i \Lambda^{\frac{1}{2}}$ where $Z_i^j \sim \mathcal{N}(0, I_{d\times d})$. Also, let $A^T A = \sum_j \lambda_i(A^T A)\mathbf{u}_j \mathbf{u}_j^T$ be the eigenvalue decomposition of $A^T A$. Then,

$$\frac{1}{n}\sum_{i=1}^{n} \Lambda^{-\frac{1}{2}} X_i^T A^T A X_i \Lambda^{-\frac{1}{2}} = \sum_{j=1}^{m} \lambda_j(A^T A)\left(\frac{1}{n}\sum_{i=1}^{n} Z_i^T \mathbf{u}_j \mathbf{u}_j^T Z_i\right). \qquad (28)$$

Also, note that $Z_i^T \mathbf{u}_j \sim N(0, I_{d\times d})$. Hence, using the standard Gaussian concentration result (see Lemma 11) along with the assumption that $n \geq Cd\log d$, we have:

$$\lambda_{\max}\left(\frac{1}{n}\sum_{i=1}^{n} \Lambda^{-\frac{1}{2}} X_i^T A^T A X_i \Lambda^{-\frac{1}{2}}\right) \leq 2\operatorname{tr}(A^T A). \qquad (29)$$

We now consider RHS of (27). Note that,

$$\frac{1}{n}\sum_{i=1}^{n} \Lambda^{-\frac{1}{2}} X_i^T A^T B \eta_i \sim \mathcal{N}(0, \beta I_{d\times d}), \qquad (30)$$

where $\beta^2 = \frac{1}{n^2}\sum_{i=1}^{n}\|A^T B \boldsymbol{\eta}_i\|_2^2 = \frac{1}{n^2}\operatorname{tr}(\Sigma_*^{\frac{1}{2}} B^T A A^T B \Sigma_*^{\frac{1}{2}} \sum_{i=1}^{n} \widetilde{\boldsymbol{\eta}}_i \widetilde{\boldsymbol{\eta}}_i^T)$, where $\widetilde{\boldsymbol{\eta}}_i \sim \mathcal{N}(0, I_{m\times m})$. Here again, using Lemma 11 we have (w.p. $\geq 1 - 1/n^{10}$):

$$\beta^2 \geq \frac{1}{2n}\operatorname{tr}(B^T A A^T B \Sigma_*).$$

Hence, w.p. $\geq 1 - 1/n^{10} - \exp(-d)$, we have:

$$\left\|\frac{1}{n}\sum_{i=1}^{n} \Lambda^{-\frac{1}{2}} X_i^T A^T \eta_i\right\|_2 \geq \frac{\sqrt{d}}{2\sqrt{n}}\sqrt{\operatorname{tr}(B^T A A^T B \Sigma_*)}. \qquad (31)$$

We obtain the following by combining (27), (29), and (31):

$$\mathbb{E}_{X\sim\mathcal{P}_X}[\|X(\widehat{\mathbf{w}} - \mathbf{w}_*)\|_2^2] \geq \frac{d}{16n} \cdot \frac{m\operatorname{tr}(B^T A A^T B \Sigma_*)}{\operatorname{tr}(A^T A)^2}. \qquad (32)$$

Note that the "$m$" term on RHS appears as $\mathbb{E}_{X\sim\mathcal{P}_X}[\|X(\widehat{\mathbf{w}} - \mathbf{w}_*)\|_2^2] = m\|\Lambda^{\frac{1}{2}}(\widehat{\mathbf{w}} - \mathbf{w}_*)\|_2^2$.

Lemma now follows by using $A = B = I_{m\times m}$ for OLS and $A = B = \Sigma_*^{-\frac{1}{2}}$ for MLE. $\qquad \square$

**Lemma 16.** *Let* $\mathbf{y}_i, X_i, \boldsymbol{\eta}_i, \mathbf{w}_*$ *be as defined in Theorem 15 and let* $\mathbf{w}_t \in \mathbb{R}^d$ *be any* fixed *vector indpendent of* $(X_i, \boldsymbol{\eta}_i)$*. Also, let* $\widehat{\Sigma}_t = \frac{1}{n}\sum(\mathbf{y}_i - X_i\mathbf{w}_t)(\mathbf{y}_i - X_i\mathbf{w}_t)^T, \Sigma_t = \Delta + \Sigma_*$, *where,*

$$\Delta = \frac{1}{n}\sum_i X_i(\mathbf{w}_* - \mathbf{w}_t)(\mathbf{w}_* - \mathbf{w}_t)^T X_i^T,$$

*and* $X_i's$ *are independent of* $\mathbf{w}_t$*. Then, w.p.* $\geq 1 - \exp(-C\cdot n)$*, the following holds* $\forall \mathbf{v} \in \mathbb{R}^d$*:*

$$\frac{1}{2}\cdot\mathbf{v}^T\Sigma_t\mathbf{v} \leq \mathbf{v}^T\widehat{\Sigma}_t\mathbf{v} \leq 2\cdot\mathbf{v}^T\Sigma_t\mathbf{v}.$$

*Proof.* Let $\mathbf{v} \in \mathbb{R}^m$ be any vector. Also, $\Sigma_t = \Delta + \Sigma_* \succ 0$ as $\Sigma_* \succ 0$. Hence,

$$\mathbf{v}^T\widehat{\Sigma}_t\mathbf{v} = \mathbf{v}^T\Sigma_t^{\frac{1}{2}}\left(\sum_i \mathbf{z}_i\mathbf{z}_i^T\right)\Sigma_t^{\frac{1}{2}}\mathbf{v},$$

where $\mathbf{z}_i = \Sigma_t^{-\frac{1}{2}} X_i(\mathbf{w}_* - \mathbf{w}_t) + \Sigma_t^{-\frac{1}{2}}\boldsymbol{\eta}_i$ is an "uncentered" Gaussian vector and hence, $\|\mathbf{z}_i\|_{\psi_2} \leq C$ for a global constant $C > 0$. Also, $\mathbb{E}[\mathbf{z}_i\mathbf{z}_i^T] = I$. Lemma now follows using standard Gaussian concentration similar to Lemma 10. $\qquad \square$

**Claim 17.** *Assume the notation of Theorem 2. Then the following holds:*

$$\frac{1}{\lambda_{\min}^*} \le \|\Sigma_*\|_2.$$

*Proof.* Let $\Sigma_* = \sum_j \lambda_j(\Sigma_*) \mathbf{u}_j \mathbf{u}_j^T$ be the eigenvalue decomposition of $\Sigma_*$. Then, we have:

$$\lambda_{\min}^* = \min_{\mathbf{v}, \|\mathbf{v}\|=1} \mathbf{v}^T \mathbb{E}_{X \sim \mathcal{P}_X} [\Sigma_X^{-\frac{1}{2}} X^T \Sigma_*^{-1} X \Sigma_X^{-\frac{1}{2}}] \mathbf{v},$$

$$\ge \min_{\mathbf{v}, \|\mathbf{v}\|=1} \sum_j \frac{1}{\lambda_j(\Sigma_*)} \mathbf{v}^T \mathbb{E}_{X \sim \mathcal{P}_X} [\Sigma_X^{-\frac{1}{2}} X^T \mathbf{u}_j \mathbf{u}_j^T X \Sigma_X^{-\frac{1}{2}}] \mathbf{v}$$

$$\ge \frac{1}{\|\Sigma_*\|_2} \min_{\mathbf{v}, \|\mathbf{v}\|=1} \mathbf{v}^T \mathbb{E}_{X \sim \mathcal{P}_X} [\Sigma_X^{-\frac{1}{2}} X^T X \Sigma_X^{-\frac{1}{2}}] \mathbf{v} = \frac{1}{\|\Sigma_*\|_2}.$$

Hence proved. $\square$

**Claim 18.** *Assume the notation of Section 2.2. Then, the following holds:*

$$\Sigma_{X*} = \mathbb{E}_{X \sim \mathcal{P}_X} [\Sigma_X^{-\frac{1}{2}} X^T \Sigma_*^{-1} X \Sigma_X^{-\frac{1}{2}}] = \frac{\operatorname{tr}(\Sigma_R \Sigma_*^{-1})}{\operatorname{tr}(\Sigma_R)} \cdot I_{d \times d}, \text{ where } \Sigma_X = \operatorname{tr}(\Sigma_R) \cdot \Lambda.$$

*Proof.*

$$\Sigma_X = \mathbb{E}_{X \sim \mathcal{P}_X}[X^T X] = \Lambda^{\frac{1}{2}} \cdot \mathbb{E}_{Z_{ij} \sim \mathcal{N}(0,1)}[Z^T \Sigma_R Z] \cdot \Lambda^{\frac{1}{2}} = \operatorname{tr}(\Sigma_R) \cdot \Lambda,$$

$$\Sigma_{X*} = \mathbb{E}_{X \sim \mathcal{P}_X} [\Sigma_X^{-\frac{1}{2}} X^T \Sigma_*^{-1} X \Sigma_X^{-\frac{1}{2}}] = \frac{1}{\operatorname{tr}(\Sigma_R)} \mathbb{E}_{Z_{ij} \sim \mathcal{N}(0,1)}[Z^T \Sigma_R^{\frac{1}{2}} \Sigma_*^{-1} \Sigma_R^{\frac{1}{2}} Z] = \frac{\operatorname{tr}(\Sigma_R \Sigma_*^{-1})}{\operatorname{tr}(\Sigma_R)} \cdot I_{d \times d}.$$

$\square$

**Corollary 19** (Result for Pooled Model, Gaussian Data, Dependent Rows)**.** *Let $X_i$ be as defined above. Let $n \ge C(m+d) \log(m+d)$. Then the followings holds (w.p. $\ge 1 - T/n^{10}$):*

$$\frac{C'd}{n} \cdot \frac{m \cdot \operatorname{tr}(\Sigma_R \Sigma_*)}{\operatorname{tr}(\Sigma_R)^2} \le \mathbb{E}_{X \sim \mathcal{P}_X}[\|X(\mathbf{w}_{OLS} - \mathbf{w}_*)\|_2^2] \le \frac{Cd \log n}{n} \cdot \frac{m \cdot \operatorname{tr}(\Sigma_R \Sigma_*)}{\operatorname{tr}(\Sigma_R)^2},$$

$$\frac{C'd}{n} \cdot \frac{m}{\operatorname{tr}(\Sigma_R \Sigma_*^{-1})} \le \mathbb{E}_{X \sim \mathcal{P}_X}[\|X(\mathbf{w}_{MLE} - \mathbf{w}_*)\|_2^2] \le \frac{Cd \log n}{n} \cdot \frac{m}{\operatorname{tr}(\Sigma_R \Sigma_*^{-1})},$$

$$\mathbb{E}_{X \sim \mathcal{P}_X}[\|X(\mathbf{w}_T - \mathbf{w}_*)\|_2^2] \le \frac{8Cd \log n}{n} \cdot \frac{m}{\operatorname{tr}(\Sigma_R \Sigma_*^{-1})} + \epsilon,$$

*where, $\mathbf{w}_T$ is the output of Algorithm 1 with $T = \log \frac{1}{\epsilon}$.*

# D   Proof of Claims from Section 3

*Proof of Theorem 7.* Let $\widetilde{X}_i \in \mathbb{R}^{m \times m \cdot d}$ be defined as:

$$\widetilde{X}_i = \begin{bmatrix} X_i^1 & 0 & \cdots & 0 \\ 0 & X_i^2 & \cdots & 0 \\ \vdots & \vdots & \vdots & \vdots \\ 0 & 0 & \cdots & X_i^m \end{bmatrix}. \tag{33}$$

Also, let $\mathbf{w}_* = \operatorname{vec}(W_*) \in \mathbb{R}^{md \times 1}$ and similarly, $\mathbf{w}_t = \operatorname{vec}(W_t), \forall t$.

Then, the observations $\mathbf{y}_i$ can be re-written as:

$$\mathbf{y}_i = \widetilde{X}_i \mathbf{w}_* + \boldsymbol{\eta}_i.$$

Similarly, we can rewrite the Step 4 in Algorithm 2 as:

$$\operatorname{vec}(W_{t+1}) = \mathbf{w}_{t+1} = \arg \min_{\mathbf{w} \in \mathbb{R}^{m \cdot d}} \frac{1}{n} \sum_{i=1}^n \|\widehat{\Sigma}_t^{-\frac{1}{2}} (\widetilde{X}_i \mathbf{w} - \mathbf{y}_i)\|_2^2.$$

That is, the above problem is a special case of the problem discussed in Section 2. First part of the theorem now follows using Lemma 3.

We now consider the second part of the theorem. Using the above given notation, we have:

$$\mathbb{E}_X \left[ \|\Sigma_*^{-\frac{1}{2}}(X \bullet W_T - X \bullet W_*)\|_2^2 \right] = \mathbb{E}_X \left[ \sum_{j,k} (\Sigma_*^{-1})_{jk} \left\langle W_T^j - W_*^j, X^j \right\rangle \left\langle W_T^k - W_*^k, X^k \right\rangle \right],$$

$$\stackrel{\zeta_1}{=} \mathbb{E}_X \left[ \sum_j (\Sigma_*^{-1})_{jj} \left\langle W_T^j - W_*^j, X^j \right\rangle^2 \right], \qquad (34)$$

where $\zeta_1$ follows from the fact that $X^j$ and $X^k$ are independent 0-mean vectors. Theorem now follows by using the above observation with the first part of the theorem. □

**Claim 20.** *Assume hypothesis and notation of Theorem 7. Then, we have:* $(\Sigma_*^{-1})_{jj} \geq \frac{1}{(\Sigma_*)_{jj}} \, \forall \, j.$

*Proof.* Let $\Sigma_* = \sum_{k=1}^m \lambda_k(\Sigma_*) \mathbf{u}_k \mathbf{u}_k^T$ be the eigenvalue decomposition of $\Sigma_*$. Now,

$$1 = \sum_{k=1}^m (\mathbf{e}_j^T \mathbf{u}_k)^2 = \sum_{k=1}^m \frac{1}{\sqrt{\lambda_k(\Sigma_*)}} \cdot \sqrt{\lambda_k(\Sigma_*)} (\mathbf{e}_j^T \mathbf{u}_k)^2 \leq (\Sigma_*^{-1})_{jj} (\Sigma_*)_{jj},$$

where the last inequality follows using Cauchy-Schwarz inequality. Hence, $(\Sigma_*^{-1})_{jj} \geq \frac{1}{(\Sigma_*)_{jj}}$. Moreover, equality holds only when $\Sigma_*$ is a diagonal matrix. □

## Footnotes

[2]For simplicity, we ignore a technicality regarding assuming that $\mathcal{S}_{t+1}$ is a fixed set. The assumption can be easily removed by taking a union bound over all sets of size $3\widetilde{s}$.