[Reviews · NeurIPS 2015]

Submitted by Assigned_Reviewer_1

SUMMARY:

This paper studies the effect of noise correlation in some models of multi-output regression. It argues that a method that does not benefit from the correlation, such as Ordinary Least Squares (OLS), may perform much worse than a method that does, such as Maximum Likelihood Estimation (MLE). For certain linear models (Pooled model and Seemingly Unrelated Regression), which are studied in the paper, the MLE estimator requires the joint optimization of the covariance and regression weights. This is a non-convex problem. Alternative Minimization (AltMin) algorithm is an approach to solve the problem by iteratively optimizing the covariance and the weights. This is not guaranteed to find the global optimum.

The contribution of the paper is a finite-sample study of AltMin and MLE and showing the effect of correlation. The results indicate that both MLE and AltMin can in fact benefit from the correlation, but OLS cannot. For certain examples, the difference can be significant. The paper provides lower bound for OLS and MLE (for the Pooled model) that up to a logarithmic factor in the number of samples is the same as the upper bound. This indicates that the upper bound of AltMin, which is the same as MLE's, is indeed tight.

The paper has some empirical studies too.

EVALUATION:

I believe this is a reasonably good paper. It provides intuition, has theory, and performs simple experiments that validate the theory. I have a few comments:

- The eps in the experiments is very small, indicating highly correlated outputs. Would the difference between MLE/AltMin and OLS be still significant when eps is much larger (maybe in the order of 0.1-0.5)? The theory suggests that the behaviour is O(eps), but what about the effect of other constants, which might be different between OLS and AltMin/MLE?

- The paper assumes that the noise model is Gaussian. How crucial is this for the conclusions of the paper? It appears that the only step that requires this Gaussian assumption is Lemma 11, which is Corollary 5.35 of [28]. It seems that one can safely use results of Section 5.3.2 of [28] to extend the results to beyond Gaussian noise models. Is this correct?

- Some typos in the proof of Theorem 15 (p13):

+ L650: The summation over i is missing in the RHS. + L656: zeta_1 is on top of the inequality symbol, but is never referred to. + L663: It seems that a log n term is missing after the equality.

- The proof of Lemma 3, L710: The inequality indicated by zeta_1 holds with a probability at least 1 - exp(-cn), but it is not stated and taken care of.

===== UPDATE: Thank you. You addressed my concerns.
Summary: I believe this is a reasonably good paper. It provides intuition, has theory, and performs simple experiments that validate the theory.

Submitted by Assigned_Reviewer_2

The paper considers the problem of multi-variable regression. The paper provides analysis of ordinary least squares and the MLE estimators for the problem and provides a characterization of when the MLE estimate is better than the OLS estimator. Further since the MLE estimator is solution to a non-convex optimization problem, for pooled model and the seemingly unrelated regression problems the paper gives finite sample analysis for alternating minimization method that can be efficiently implemented. For these problems matching upper and lower bounds are provided for OSL, MLE and upper bounds for alternative minimization approach. The analysis clearly

points out when the MLE/alternative minimization approach provides improvements for these problems.

Overall the paper is well written and easy to follow. The proofs are straightforward and simple. As far as I checked the proofs seem fine. The analysis for MLE VS that for OSL seems straightforward. What really sells this work is that the theorems are proved for the AM based approach.

However I do have a few concerns. 1.

The models considered are all well specified models but the applications listed as motivation are more machine learning type problems where well specified assumption is almost never true. The analysis is very heavily dependent on the well specifiedness.

2. The significance of the results is not clear. On the experiments side, synthetic data is used. On the theory side, that alternative minimization (and MLE) yields better performance is clear but how significant is this in terms of applications is not clear. Can you provide concrete examples of applications where the gain is significant.

Seems like for the difference between MLE and OSL to be large enough one needs the dimensionality to be large enough at the very least.

3. Finally the results seem (although only a bit) out of scope for NIPS seems more classical stats econometrics type work.

Overall I would like to see the paper accepted if there is space at NIPS.
Summary: The results are certainly interesting but the significance of the results especially w.r.t. scope of NIPS is not clear.

Submitted by Assigned_Reviewer_3

The paper considers regression problem with multiple vector valued outputs, that are related either via sharing a common noise vector, or the same coefficient vectors, or some other attributes. This is an interesting theoretical and practical question.

The paper considers two popular models for such problems. The first is the pooled model where the coefficient vector is shared across the various outcomes. The second is the SUR model which shares the noise vector across the dimensions. In both of these problems it is possible to simply ignore the additional information that can be extracted and perform the ordinary least squares regression (OLS). It can be shown that MLE outperforms this procedure in these settings. However, MLE requires solving a non-convex problem. Therefore, the authors study alternating minimization, and under these models show that its performance is within universal constant factors of MLE.

1. lines 190-200: Is the repeated use of fresh samples the reason for the additional logarithmic factor in theorem 1? It would be

interesting to see why the logarithmic factor pops up, since the lower bounds do not seem to have it.

2. It would be interesting to apply these results to some real world data.

3. The two models considered are in some sense restrictive. It would be nice if the authors can comment on

what the reason for Alt-Min performing well under these models is. In particular, are there generic conditions under which Alt-Min can be shown to be within universal constant factors of MLE for regression problems?

Summary: The paper shows that under two popular models for regression problems with vector valued outputs, Alternating Minimization can match the performance of MLE estimation up to universal constants. This is an interesting result.

Author Feedback
Author rebuttal: We first respond to common concerns and then address points raised by individual reviewers.

COMMON CONCERNS
Work is classical stats/econometrics type: We want to emphasize that the novelty in our work is two-fold: a) provide provable *computational* guarantees for solving a popular MLE problem which is non-convex, b) provide finite sample bounds. Both of these are missing from the classical statistics literature and in fact these properties (computational complexity and finite sample bounds) have typically got more attention in the ML community. Moreover, our model is frequently used in the multitask learning domain (see cited papers). Hence, we believe the work is easily within the scope of NIPS.

Work relies on well-specified model and Gaussian error: Our analysis is more general and is couched in terms of sub-Gaussian norms of random vectors. Model assumptions are, by definition, necessary for interpreting an estimator as an MLE (likelihood needs a model). However, if one only cares for solving the following natural non-convex problem: min_{\Sigma, w} \sum_i \|\Sigma^{-1/2}(X_i w-y_i)\|^2-\log|\Sigma|, even though the noise in the model is *not Gaussian*. Then also our results would hold (with different constants) and provide improvements (in terms of error bounds) over OLS. But naturally, we cannot compare against the true MLE of that noise model, because the above given problem itself is not the MLE estimator.

Real world data not used in experiments: We wish to point out that there are textbook examples of real world data where using even a few iterations of AltMin reduces standard errors by several factors over OLS solution. See, for example, results for pooled model on a real world data set given in Table 10.1 (pp. 302-303) of Greene's book that we cited in the paper. The question this research addressed is, in fact, motivated by a desire to explain what practitioners regularly observe.

REVIEWER 1
For larger \epsilon as well (we tried up to \epsilon=0.9 in our experiments), we observe that MLE/AltMin performs better than OLS. As we observe gains for reasonably large \epsilon, it indicates that the constants for MLE/AltMin and OLS are quite similar.
Thanks for pointing out the typos, we will fix it in the next draft.

REVIEWER 2
We have responded to most of your concerns in the common concerns above.

REVIEWER 3
lines 190-200: Yes extra log factor arises because of use of fresh samples.

REVIEWER 4
Thanks for your positive review.

REVIEWER 5
See reply to common concerns above. log n factor appears due to fresh samples at each step and it is not clear if that can be avoided as re-using samples introduces fairly intricate dependencies between the iterates and samples.

REVIEWER 6
Thanks for your positive review.